# Optimization of Mapping Tools and Investigation of Ribosomal RNA Influence for Data-Driven Gene Expression Analysis in Complex Microbiomes

**DOI:** 10.3390/microorganisms13050995

**Published:** 2025-04-26

**Authors:** Ryo Mameda, Hidemasa Bono

**Affiliations:** 1Graduate School of Integrated Sciences for Life, Hiroshima University, 3-10-23 Kagamiyama, Higashi-Hiroshima 739-0046, Japan; 2Genome Editing Innovation Center, Hiroshima University, 3-10-23 Kagamiyama, Higashi-Hiroshima 739-0046, Japan

**Keywords:** metagenomics, metatranscriptomics, NGS, gene expression, read mapping, ribosomal RNA

## Abstract

For gene expression analysis in complex microbiomes, utilizing both metagenomic and metatranscriptomic reads from the same sample enables advanced functional analysis. Due to their diversity, metagenomic contigs are often used as reference sequences instead of complete genomes. However, studies optimizing mapping strategies for both read types remain limited. In addition, although transcripts per million (TPM) is commonly used for normalization, few studies have evaluated the influence of ribosomal RNA (rRNA) in metatranscriptomic reads. This study compared Burrows–Wheeler Aligner–Maximal Exact Match (BWA-MEM) and Bowtie2 as mapping tools for metagenomic contigs. Even after optimizing Bowtie2 parameters, BWA-MEM showed higher efficiency in mapping both metagenomic and metatranscriptomic reads. Further analysis revealed that rRNA sequences contaminate predicted protein-coding regions in metagenomic contigs. When comparing TPM values across samples, contamination by rRNA led to an overestimation of TPM changes. This effect was more pronounced when the difference in rRNA content between samples was larger. These findings suggest that metatranscriptomic reads mapped to rRNA should be excluded before TPM calculations. This study highlights key factors influencing read mapping and quantification in gene expression analysis of complex microbiomes. The findings provide insights for improving analytical accuracy and advancing functional studies using both metagenomic and metatranscriptomic data.

## 1. Introduction

Complex microbiomes, such as soil microbes, aquatic microbes, and gut bacteria, are ubiquitous in the environment and play critical roles in influencing the higher organisms with which they coexist. Additionally, certain complex microbiomes, such as those in activated sludge, are utilized in industrial applications, making the elucidation of their functions highly valuable.

Metagenomic and metatranscriptomic analyses using next-generation sequencing (NGS) have been widely employed to investigate the functions of complex microbiomes. Studies targeting individual microbial species include investigations of nitrification genes in *Nitrosomonas* sp. within activated sludge [1] and the discovery of antibiotic biosynthesis genes in Actinomycetes from soil microbiomes [2]. Broader processes, such as the nitrogen cycle [3] and carbon cycle [4], have also been analyzed in soil microbiomes. These examples underscore the increasing application of metagenomic and metatranscriptomic approaches in functional microbiome research.

Reference sequences is a part of critical factor in the functional analysis of complex microbiomes. In NGS-based analyses, NGS reads must be compared against reference sequences with functional annotations. For single organisms, complete genome sequences can serve as references; however, this approach is challenging for complex environmental microbiomes due to their high diversity. Recent advancements in microbial genome sequencing have introduced methods such as metagenome-assembled genomes (MAGs) for bacteria with relative abundances of less than 1% in complex microbiomes [5] and single-amplified genomes (SAGs) obtained through single-cell analysis [6]. Notably, approximately half of the genomes in Release 220 of the Genome Taxonomy Database consist of MAGs or SAGs [7]. Despite these advances, only an estimated 2.1% of environmental bacteria have been genome-sequenced [8], highlighting the limitations of using database-registered genome data as references for species and gene function identification in complex microbiomes. In previous studies, metagenomic contigs have been used as reference sequences and allow for more comprehensive coverage of reads obtained from samples containing a large proportion of uncultured microorganisms, such as soil microbiomes [9,10]. In addition, covering both metagenomic and metatranscriptomic reads enables more advanced differential gene expression analysis [11,12]. In such analyses, mapping reads to reference sequences and quantifying gene expression using read counts are common [9,10,13]. Some studies have focused on optimizing mapping tools specifically for metagenomic analysis, and Burrows–Wheeler Aligner (BWA) and Bowtie2 have demonstrated great benchmarking performance [14,15,16]. These mapping tools have consistently demonstrated efficient performance for metagenomic reads in previous studies. Although efficient mapping tools that can accurately align both types of reads are essential, there has been a lack of comparative analyses evaluating their effectiveness for metatranscriptomic reads.

In addition to read mapping, methods for evaluating gene expression levels must also be considered. Normalization is considered essential for comparing samples, and transcripts per million (TPM) is widely used [17,18]. TPM represents the proportion of read counts for a given gene relative to the total expressed genes in a sample, normalized by gene length. As a result, TPM values can be influenced by the expression levels of other genes present in the transcriptome. However, its specific effects on gene expression analysis in complex microbiomes have not been thoroughly investigated. This study focused on two key aspects of the gene expression analysis of complex microbiomes: read mapping and the influence of ribosomal RNA (rRNA) on TPM calculation. For read mapping, the widely used tools Burrows–Wheeler Aligner–Maximal Exact Matches (BWA-MEM) and Bowtie2 were compared. Additionally, contamination of rRNA sequences within the protein-coding sequences of metagenome contigs was detected, demonstrating its potential impact on TPM calculation. The analysis was conducted using metagenomic and metatranscriptomic reads obtained from the same samples available in the National Center for Biotechnology Information (NCBI) database.

## 2. Materials and Methods

The several shell scripts used in this research are available in the GitHub repository at https://github.com/RyoMameda/workflow (accessed on 22 April 2025). The development into a workflow language will be released.

### 2.1. Computational Resources

This analysis was conducted using a system equipped with 64 GB RAM and an Apple M1 Max chip, operating on macOS Sequoia. The binning step was performed on a Linux environment using the NIG supercomputer at the ROIS National Institute of Genetics.

### 2.2. Optimization of Mapping Tools

The analysis utilized 56 short-read datasets of soil microbiomes. Metagenomic and metatranscriptomic reads were obtained from 24 samples each, as described in reference studies [9,10,13,19] (Appendix A). Two reference studies mentioned rRNA depletion before sequencing [9,13]. Over 95% of bases in the raw reads had a quality score of Q20 or higher, as confirmed by quality control described below. FASTQ files for these reads were retrieved from the NCBI Sequence Read Archive (SRA) database using the SRA Toolkit [20] (v3.0.10) with the prefetch and fasterq-dump commands. Quality control and trimming were performed using fastp [21] (v0.23.4) with the parameters -q 20 -t 1 -T 1. Trimmed metagenomic reads were assembled into contigs using MEGAHIT [22,23] (v1.2.6) with default parameters. Protein-coding sequences were predicted from metagenomic contigs using Prodigal [24] (v2.6.3) with the -p meta parameter. Trimmed metagenomic and metatranscriptomic reads were mapped to the predicted protein-coding sequences using BWA MEM [25] (v0.7.17) or Bowtie2 [26,27,28] (v2.5.1). Sequence Alignment/Map (SAM) files generated during mapping were converted to Binary Alignment/Map (BAM) format using SAMtools [29] (v1.17) with the sort command, and mapping statistics were analyzed with the flagstat command. Mapping quality was analyzed using Qualimap2 [30] (v2.3).

### 2.3. Gene Expression Analysis

Gene annotation was assigned to the predicted protein-coding sequences as follows. Predicted nucleotide sequences of metagenomic contigs were screened against rRNA sequences from NCBI [31] using BLASTN [32] (v2.15.0) with an E-value threshold of 0.1. The remaining protein sequences were annotated by querying against the Swiss-Prot database of well-characterized proteins in UniProt [33] using DIAMOND [34] (v2.0.15) with BLASTP and an E-value threshold of 0.1. Sequences without hits in Swiss-Prot were further annotated using protein domain information from Pfam [35] via HMMER [36] (v3.3.2) with an E-value threshold of 0.1, and were parallelized using GNU Parallel (v20230922) [37]. Predicted protein sequences were annotated based on the highest-ranking hits from the BLASTP and HMMER searches, with sequences lacking significant matches designated as hypothetical proteins. Using the annotations and BAM files from the previous steps, read counts for each sequence were quantified with the featureCounts command in Subread [38] (v2.0.6). Both metagenomic and metatranscriptomic read counts were normalized using the following equation. For a given contig *t*, let *T*_t_ represent the metatranscriptomic read count, and *L*_t_ the contig length. Transcripts per million (*TPM*) was calculated accordingly (1).(1)$$TPM=\frac{\left(\frac{T_{t}}{L_{t}}10^{3}\right)}{\sum_{t}\left(\frac{T_{t}}{L_{t}}10^{3}\right)}10^{6}$$

## 3. Results and Discussion

### 3.1. Mapping Rates to Metagenomic Contigs

The analytical pipeline, encompassing processes from contig construction to gene expression quantification, was developed and integrated (Figure 1).

First, both metagenomic and metatranscriptomic reads were mapped on metagenomic contigs constructed by MEGAHIT with BWA-MEM or Bowtie2 under default parameters. Overall, BWA-MEM achieved a higher mapping rate than Bowtie2 (sensitive preset) for both read types (Figure 2). Although previous studies reported that BWA-MEM achieves higher mapping rates for metagenomic contigs and those mapping rates were similar as the results in this study [14], it was hypothesized that Bowtie2 parameters might not have been fully optimized. While BWA-MEM allows local alignment and has a default seed length of 19, the Bowtie2-sensitive preset is end-to-end mode. To bring Bowtie2 settings closer to BWA-MEM, local or very-sensitive-local preset with a seed length of 19 (-L 19) was used. This local alignment setting significantly improved mapping rates (Figure 2). However, modifying Bowtie2’s mismatch penalty to match BWA-MEM (setting it to 4) did not further improve mapping rates (Appendix A). Given these optimizations, the change in mapping rate was expected to affect mapping quality. Contrary to expectations, parameter changes did not result in noticeable differences in mapping quality (Appendix A). These results indicate that while optimizing Bowtie2 parameters can improve mapping rates, BWA-MEM remains a more efficient tool for mapping both metagenomic and metatranscriptomic reads. This finding represents an important insight for the gene expression analysis of complex microbiomes utilizing both types of reads. Both Bowtie2 and BWA-MEM utilize FM-index [39] and Burrows–Wheeler transform [40], but they differ in alignment features: BWA-MEM bases alignment on the number of mismatches, whereas Bowtie2 relies on an alignment score [41]. Whether this difference accounts for the observed variation in mapping rates remains unclear, requiring further investigation.

### 3.2. Influence of rRNA on TPM Calculation

Predicted protein sequences derived from the metagenomic contigs were found to include rRNA sequences due to chance translational frames. To address this, BLASTN searches were conducted against rRNA sequence databases from NCBI. For instance, in the metagenome contigs assembled from SRR24888648, only 0.09% of the predicted protein-coding sequences were identified as rRNA. However, an analysis of mapped read counts by BWA-MEM revealed that 0.16% (155,043 of 98,847,988) of metagenomic reads and 36.0% (23,079,523 of 64,026,082) of metatranscriptomic reads (SRR24887388) mapped to rRNA sequences. Although in vitro rRNA removal was performed in the referenced study [9], residual rRNA contamination persisted in metatranscriptomic reads, affecting predicted protein-coding sequences. In contrast, for the predicted protein-coding contigs of SRR22507541, in which rRNA was not removed in vitro [19], 95.1% (34,017,387 of 35,774,766) of the metatranscriptomic reads (SRR22506317) were mapped to rRNA. While rRNA removal kits are known to be effective, residual rRNA sequences are still commonly observed [42,43]. This study further revealed that such rRNA sequences can also contaminate predicted protein-coding sequences, highlighting an additional source of potential bias in downstream analyses.

The impact of residual rRNA on TPM calculation was investigated. In a reference dataset [9] where rRNA was removed in vitro, the total TPM value of rRNA-mapped reads was 305,514 and 536,616 in SRR24888495 and SRR24887388, respectively. For genes commonly expressed in both samples, TPM values were calculated with and without including rRNA, and changes were expressed as log_10_ fold differences. The results showed that TPM values calculated with rRNA included were slightly but consistently higher compared to those calculated with rRNA excluded (Figure 3A). In contrast, when comparing SRR24887388 with SRR24887267 (total rRNA TPM values are 536,616 and 513,379, respectively), the variability in gene expression between samples was minimal (Figure 3B). To further assess the impact, an extreme case was analyzed using data from another reference study [19] in which metatranscriptomic reads were obtained without in vitro rRNA removal. TPM values were calculated both with rRNA included and after the removal of rRNA. In SRR22506317 and SRR22506321, total rRNA TPM values were 970,933 and 980,113, respectively. When comparing gene expression ratios between the two samples, the difference became more noticeable when rRNA was excluded from only one of the samples during TPM calculation (Figure 3C). These findings indicate that larger differences in rRNA contamination between samples result in greater discrepancies in gene expression variability. Therefore, consistent rRNA removal across samples is essential during data processing to ensure accurate expression analysis.

These results demonstrate that when the residual rRNA varies across samples, calculating TPM with rRNA included can lead to inconsistencies in the interpretation of differential gene expression analysis. By excluding rRNA read counts from TPM calculation, such inter-sample noise can be minimized, allowing for a more accurate assessment of gene expression variation. Although the potential impact of rRNA on TPM has been previously mentioned in single organisms [44], this study showed that differences in calculation can lead to measurable variations. Alternatively, tools such as SortMeRNA [45] can be employed to remove rRNA sequences in silico from metatranscriptomic reads prior to analysis. Further investigation is needed to evaluate the effectiveness of these approaches in detail. Additionally, TPM represents the relative expression level normalized across samples. In complex microbiomes, where genes originate from various microbial species, normalization methods using both read counts and gene length might improve the accuracy of gene-specific expression analysis [46]. Furthermore, incorporating the gene copy number from the metagenome into the analysis has been suggested to further refine the evaluation of gene expression dynamics [12]. Improving cross-sample quantification accuracy may require approaches such as incorporating internal standards during NGS read acquisition [47]. Furthermore, more efficient methods for extracting DNA and RNA from environmental microbiome samples need to be explored [48,49,50]. Combining such advanced sample preparation techniques with this study could further enhance the precision and scalability of quantitative gene expression analysis in complex microbiomes.

## 4. Conclusions

This study focused on the gene expression analysis of complex microbiomes using NGS. It is well established that mapping both metagenomic and metatranscriptomic reads obtained from the same sample to metagenomic contigs enables advanced expression analysis. Evaluation of mapping tools in this study revealed that BWA-MEM efficiently maps both types of reads. Additionally, TPM is commonly used for normalizing mapped read counts during gene expression analysis. However, rRNA contamination was found in predicted protein sequences derived from metagenomic contigs, and metatranscriptomic reads often contain residual rRNA that cannot be completely removed in vitro. These factors were shown to potentially influence TPM calculation. Applying these findings to the functional analysis of complex microbiomes can contribute to more accurate and advanced gene expression analysis.

## Figures and Tables

**Figure 1 microorganisms-13-00995-f001:**
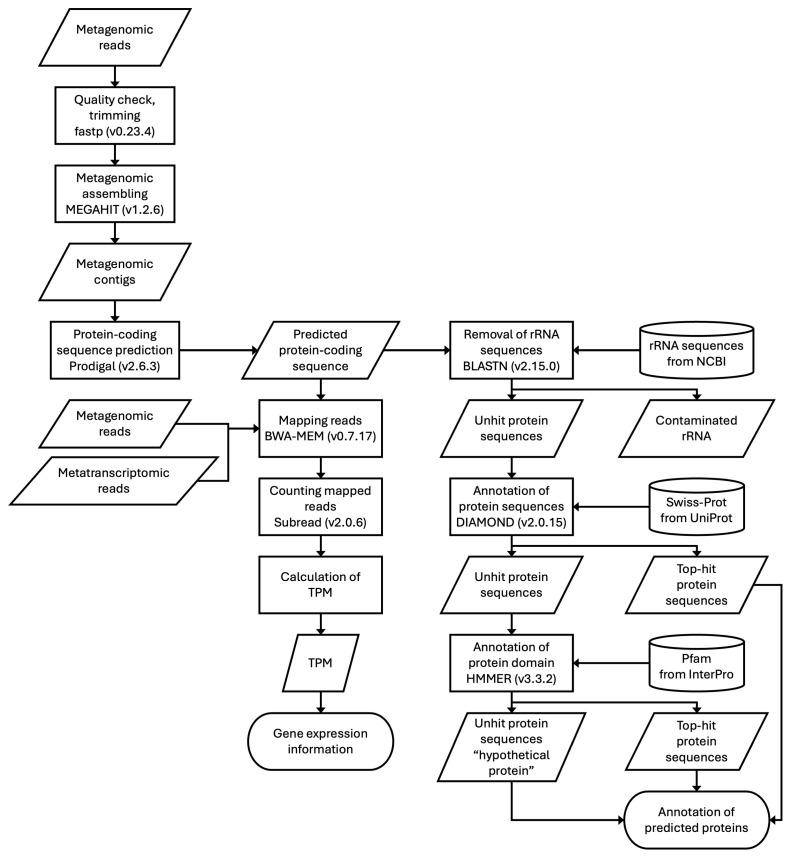
Data-driven analytical pipeline for gene expression analysis in complex microbiomes. Features of DNA sequences and read data are shown in rhombuses, processing methods are indicated in rectangles, datasets used as annotation references are placed in cylinders, and the final output data are presented in rounded rectangles.

**Figure 2 microorganisms-13-00995-f002:**
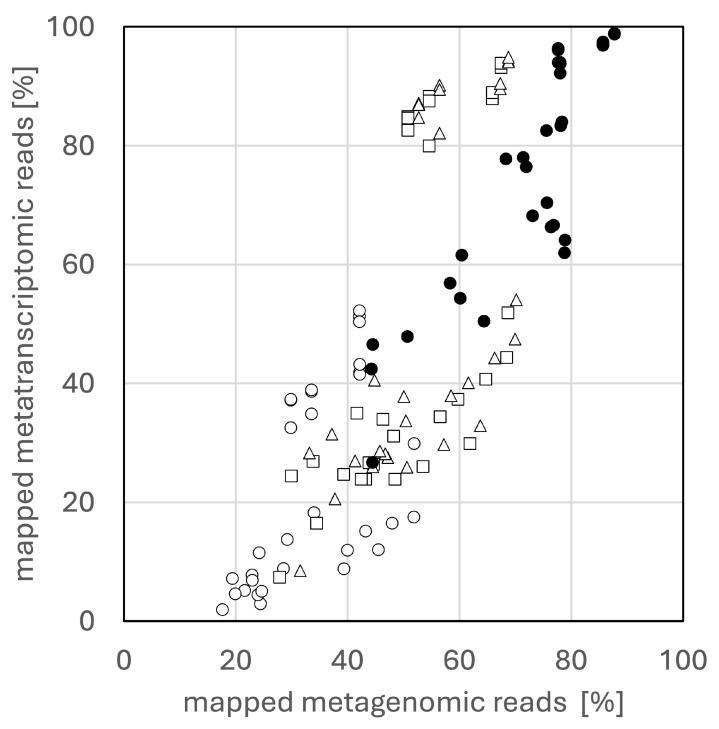
Mapping rates of metagenomic and metatranscriptomic reads. Reads were mapped using BWA-MEM (filled circle) or Bowtie2 (outlined shapes), whose setting was sensitive (circles), local -L 19 (squares), or very-sensitive-local -L 19 (triangles).

**Figure 3 microorganisms-13-00995-f003:**
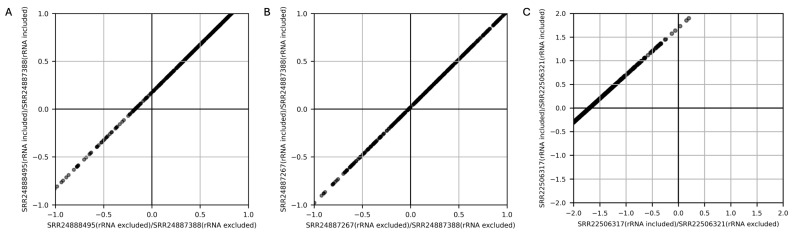
Log-fold change in TPM between samples. A random subset of 500 expressed genes were plotted. A comparison was conducted between (**A**) SRR24888495 and SRR24887388, (**B**) SRR24887267 and SRR24887388, (**C**) SRR22506317 and SRR22506321 to evaluate the impact of rRNA removal.

## Data Availability

The shell scripts used in this research are available in the GitHub repository at https://github.com/RyoMameda/workflow (accessed on 22 April 2025).

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
