# Peer review of "Optimization of Mapping Tools and Investigation of Ribosomal RNA Influence for Data-Driven Gene Expression Analysis in Complex Microbiomes"

_microorganisms, 2025, doi:10.3390/microorganisms13050995_

Round 1
Reviewer 1 Report
Comments and Suggestions for Authors
This study optimized mapping tools and investigated ribosomal RNA influence for data-driven gene expression analysis in complex microbiomes. Below are points need to be improved:
Abstract: Line 17 "TPM" and line 19 "BWA-MEM" need to be spelled full.
Introduction: Line 65 "BWA" and line 79 "BWA-MEM" need to be spelled full when first mentioned.
Materials and Methods: Materials and Methods need to be more detailed to be able to repeat.
Line 93 The sources and features of dataset used in the analysis need to be listed. Have the samples rRNA been depleted before sequencing? Which methods have been used for rRNA depletion? Supplementary tables couldn't be found.
Results: Figure 1 belongs to Methods section. Noticed Figure 1 uses different shapes of text boxes. Are there any particular meanings or groups for the specific shapes?
Can't find supplementary materials. Not uploaded.
Suggest adding a list of abbreviation at the end of manuscript before References.
Comments on the Quality of English LanguageThe manuscript need to be written clearer.
Author Response
Comments 1: Abstract: Line 17 "TPM" and line 19 "BWA-MEM" need to be spelled full. Introduction: Line 65 "BWA" and line 79 "BWA-MEM" need to be spelled full when first mentioned.
Response 1: Agree. We have added descriptions in the Abstract and Introduction sections, defining TPM as "Transcripts Per Million" in line 15 and BWA-MEM as "Burrows-Wheeler Aligner Maximal Exact Matches" in line 17, 77, 92.
Comments 2: Line 93 The sources and features of dataset used in the analysis need to be listed. Have the samples rRNA been depleted before sequencing? Which methods have been used for rRNA depletion? Supplementary tables couldn't be found.
Response 2: Thank you for pointing this out. We have uploaded supplementary materials listing the features of the datasets used in this study. Additionally, we clarified which reference studies involved rRNA depletion by stating at line 110: “Two reference studies mentioned rRNA depletion before sequencing [9,13].”
Comments 3: Results: Figure 1 belongs to Methods section. Noticed Figure 1 uses different shapes of text boxes. Are there any particular meanings or groups for the specific shapes?
Response 3: There is meaning assigned to the boxes used in the Figure 1. We revised the description in lines 147–149 as follows: “Features of DNA sequences and read data are shown in rhombuses, processing methods are indicated in rectangles, datasets used as annotation references are placed in cylinders, and the final output data are presented in rounded rectangles.”
Comments 4: Suggest adding a list of abbreviation at the end of manuscript before References.
Response 4: We added the missing abbreviations in the main text, such as ribosomal RNA (rRNA) in line 91, National Center for Biotechnology Information (NCBI) in line 96, Sequence Read Archive (SRA) in line 113, and Binary Alignment/Map (BAM) in line 121. Additionally, we provided a list of them at the end (line 267).
Reviewer 2 Report
Comments and Suggestions for Authors
This manuscript optimized the mapping tools and investigated the ribosomal RNA influence for data-driven gene expression analysis in complex microbiomes. These results provide insights for improving analytical accuracy and advancing functional studies using both metagenomic and metatranscriptomic data. However, prior to being accepted, there are some issues that should be addressed.
- The Abstract Section is not very accurate, and many key results are not included. For example, the difference became more noticeable when rRNA was excluded from only one of the samples during TPM calculation.In addition, the research background in the Abstract Sectioncan be appropriately reduced in some content.
- Lines 17, 19, TPM and BWA-MEMA were appearing for the first time and must be written in its full name.
- Lines 14, 42, Generally speaking, only the species or genus names of microorganisms need to be italicized, while other names such as phyla, classes, orders, and families do not need to be italicized. Here, Nitrosomonas is a generic name and should be italicized; However, Actinomycetes are not specific species or genus names, so there is no need to italicize them.
- 4. Lines 93, 94, Are these 56 short read datasets of soil microbiomes randomly selected? Is there still some basis? If so, please explain.
- Line 99, MEGAHIT was appearing for the first time and must be written in its full name.
- 6. Line 101, Is the description of '- p meta parameter' accurate? Please carefully verify.
- Some references are not very standardized, such as Ref. 1, 2,3, 5, 9,13, 26, 27, 34, 43, 44, the journal names should be written in full.
Author Response
Comments 1; The Abstract Section is not very accurate, and many key results are not included. For example, the difference became more noticeable when rRNA was excluded from only one of the samples during TPM calculation. In addition, the research background in the Abstract Section can be appropriately reduced in some content.
Response 1: Thank you for your constructive comments and we agree it. In the Abstract section, we have reduced in content of the research background in lines 11-14. In addition, we revised results to include that rRNA contamination can affect TPM comparison between samples in lines 22-25. We hope that the Abstraction section has become more accurate in its expression.
Comments 2: Lines 17, 19, TPM and BWA-MEMA were appearing for the first time and must be written in its full name.
Response 2: Agree. We have added descriptions in the Abstract section, defining TPM as "Transcripts Per Million" in line 15 and BWA-MEM as "Burrows-Wheeler Aligner Maximal Exact Matches" in line 17.
Comments 3: Lines 41, 42, Generally speaking, only the species or genus names of microorganisms need to be italicized, while other names such as phyla, classes, orders, and families do not need to be italicized. Here, Nitrosomonas is a generic name and should be italicized; However, Actinomycetes are not specific species or genus names, so there is no need to italicize them.
Response 3: Agree. We have changed “Nitrosomonas” to italicized style in line 53 and "Actinomycetes" to regular style in line 54.
Comments 4: Lines 93, 94, Are these 56 short read datasets of soil microbiomes randomly selected? Is there still some basis? If so, please explain.
Response 4: Thank you for pointing this out. These datasets were selected based on their base quality. Therefore, we added “Over 95% of bases in the raw reads had a quality score of Q20 or higher, as confirmed by quality control described below.” in line 110-112.
Comments 5: Line 99, MEGAHIT was appearing for the first time and must be written in its full name.
Response 5: MEGAHIT is a commonly used tool name and might be not an abbreviation.
Comments 6: Line 101, Is the description of '- p meta parameter' accurate? Please carefully verify.
Response 6: Although the “-p meta” parameter was replaced by “-p anon” in the recent version 3 of Prodigal, the “-p meta” parameter was still available in version 2 which was used in this study.
Comments 7: Some references are not very standardized, such as Ref. 1, 2,3, 5, 9,13, 26, 27, 34, 43, 44, the journal names should be written in full.
Response 7: We intended to follow the journal’s guidelines, in which abbreviated journal names should be used in the References section.
Reviewer 3 Report
Comments and Suggestions for Authors
The article titled "Optimization of Mapping Tools and Investigation of Ribosomal RNA Influence for Data-driven Gene Expression Analysis in Complex Microbiomes" presents a thorough and well-structured investigation into the critical factors affecting read mapping and quantification in gene expression analysis within complex microbiomes. The study is written in a clear and accessible manner, making it suitable for a broad academic audience. The conclusions drawn align well with the presented results, showcasing rigorous methodology and insightful analysis.
While the article is commendable in its overall presentation, one minor revision is recommended to enhance clarity for readers unfamiliar with specific terminology. In Abstract, the abbreviation "TMP" is introduced prior to providing its full definition. To improve accessibility for readers who may not be acquainted with this term, it would be beneficial to include its full name in the Abstract section. This adjustment would ensure that the text remains comprehensible to a wider audience without requiring prior familiarity with the abbreviation.
Overall, this study makes a valuable contribution to the field of microbiome research and gene expression analysis, offering practical insights into optimizing mapping tools and addressing key challenges in data-driven approaches.
Author Response
Comments 1: While the article is commendable in its overall presentation, one minor revision is recommended to enhance clarity for readers unfamiliar with specific terminology. In Abstract, the abbreviation "TMP" is introduced prior to providing its full definition. To improve accessibility for readers who may not be acquainted with this term, it would be beneficial to include its full name in the Abstract section. This adjustment would ensure that the text remains comprehensible to a wider audience without requiring prior familiarity with the abbreviation.
Response 1: Thank you for your reviewing. We agree with the minor revision which you suggested. We have added several descriptions in the Abstract section. For example, defining TPM as "Transcripts Per Million" in line 15.
Round 2
Reviewer 2 Report
Comments and Suggestions for Authors
The author has already responded to the issues I am concerned about one by one, and I have no further comments.